# Conformal Swallowing Accelerometry: Reimagining the Acquisition and Characterization of Swallowing Mechano-Acoustic Signals

**DOI:** 10.3390/s25237396

**Published:** 2025-12-04

**Authors:** Wilson Yiu Shun Lam, Elaine Kwong, Randolph Chi Kin Leung, Chak Hang Lee, Sanjaya Rai, Leo Kwan Lui

**Affiliations:** 1Department of Language Science and Technology, The Hong Kong Polytechnic University, Hong Kong SAR 999077, China; yiu-shun-wilson.lam@connect.polyu.hk (W.Y.S.L.);; 2Research Institute for Smart Ageing, The Hong Kong Polytechnic University, Hong Kong SAR 999077, China; mmrleung@polyu.edu.hk; 3Department of Mechanical Engineering, The Hong Kong Polytechnic University, Hong Kong SAR 999077, China

**Keywords:** deglutition, swallowing, swallowing acoustics, conformal swallowing accelerometry

## Abstract

**Highlights:**

This study utilized a conformal array of accelerometers to revisit the core assumptions of swallowing signal reproducibility and symmetry that justify the single-point approach to cervical auscultation and swallowing accelerometry.

**What are the main findings?**

**What is the implication of the main finding?**

**Abstract:**

(1) Background: Non-invasive instrumental measurement of swallowing acoustic signals has rested upon the assumptions of signal symmetry and reproducibility along the cervical region and has hence taken the form of single-point acquisition on optimal sites. This study aimed to (i) revisit such assumptions by adopting a conformal array of accelerometers, and hence (ii) lay the foundation for the future design of swallowing accelerometry. (2) Methods: Thirteen young healthy individuals, including eight females (mean age ± SD = 24.38 ± 0.92) and five males (mean age ± SD = 24 ± 3.74), were recruited. Swallowing mechano-acoustic signals of repeated swallowing trials were captured using conformal swallowing accelerometry. The peak intensities and frequencies as well as their respective peak times were extracted from six symmetrical and vertically aligned sites. (3) Results: Three-way ANOVAs with repeated measures suggested differences across trials and channels for both peak intensity and frequency. The additional interaction of bolus volume and repeated trials with a small effect size was also indicated in peak frequency. Intra-personal variability was indicated by coefficients of variance of the peak intensity and frequency of higher than 20%, with values varying within the 95% limits of agreement of at least 10 m/s^2^ and 100 Hz, respectively. However, intra-trial comparisons of contra-lateral peak intensity and frequency also revealed a high degree of variability, with the 95% limits of agreement up to 12 m/s^2^ and 240 Hz, respectively. On the other hand, the time points of intra-trial peak intensity and frequency showed a high degree agreement, suggesting the possibility of signal asymmetry. (4) Conclusions: The current findings not only confirmed the previous proposal of intra-personal variability but also demonstrated preliminary counterevidence to the longstanding assumption of signal symmetry. Alternatively, the use of conformal swallowing accelerometry is a promising option for the future design and implementation of non-invasive swallowing mechano-acoustic measurements.

## 1. Introduction

The non-invasive acoustic measurement of deglutition has gained popularity as an adjunctive modality for understanding and evaluating swallowing physiology in research and clinical practice over the past few decades. Such measurements usually take the form of swallowing accelerometry (SA) [1,2] or cervical auscultation (CA) [3], respectively, with an accelerometer as the tool of signal acquisition, or a listening device (e.g., a contact microphone or stethoscope). Given that the upper aerodigestive tract is seen as a set of tubes and valves [4], the basic working principle of CA/SA rests on the assumption that we can assess swallowing mechanics, hence physiology, through the sounds (for CA) or vibrations (for SA) caused by the opening and closing of various anatomical structures (e.g., the cardiac analogy hypothesis in [5]).

### 1.1. Cervical Auscultation (CA) and Swallowing Accelerometry (SA)

Although sharing similar physical measurement principles, CA and SA measure different physical properties of swallowing physiology. The listening device used in CA evaluates sound/acoustic pressure, in the units of Pascals (Pa), assumed to be caused by motions of swallowing. The sounds/acoustic signals are usually audible, i.e., within the threshold of human hearing, and hence could be acquired and perceived through a (digital) stethoscope. One can compare this technique to phonocardiography where heart sounds are acquired and perceived through a microphone or stethoscope. On the other hand, SA evaluates mechanical vibratory acceleration, in the units of meter per second squared (m/s^2^) or g-force (g), on the skin surface along the anterior-posterior/lateral/inferior-superior axes (e.g., [6]). The resulting mechano-acoustic signals, by the same principle, are believed to be caused by swallowing movements. In the context of cardiac motion measurement, this technique can be compared to seismocardiography, which assesses cardiac mechanics through chest wall vibration. Their difference has been evidenced and discussed by the recent findings from high-resolution cervical auscultation (HRCA) (e.g., [7,8,9]), where both CA and SA were used simultaneously.

### 1.2. The Nature of Swallowing Mechano-Acoustic Signals: The Know-What

So far, the nature of swallowing sounds and vibrations has been assumed with respect to findings in the 1990s. Drawing findings from simultaneous and symmetrical measurements of swallowing movements with accelerometers on each of the lateral sides of the cricoid cartilage, Takahashi and colleagues [1] concluded that swallowing mechano-acoustic signals were grossly symmetrical and reproducible on both sides of the neck, and thus a single-point acquisition with repeated trials would suffice for obtaining information about swallowing dynamics. Subsequently, there has been little work to replicate the results or revisit such proposals of symmetry and reproducibility.

Regarding the cause of swallowing sounds/vibrations, various assumptions were discussed extensively in [5], which then proposed the cardiac analogy hypothesis. The cardiac model of the hypothesis postulates that swallowing sounds arise from vibrations within closed chambers (e.g., the pharynx), and the vocal tract model assumes that the perceptual features of the sounds (e.g., frequency) is a function of the configuration of the pharynx during swallowing. Since then, there have been mixed findings with respect to this hypothesis. With the use of HRCA, Kurosu and colleagues [8] found the associations between several acoustic features with various movements during swallowing (e.g., laryngeal vestibule closure and re-opening, upper esophageal sphincter opening, and base of tongue and posterior pharyngeal wall contact). Subsequently, He and colleagues [10] also found strong correlations between HRCA acoustic features and hyoid bone displacement from videofluoroscopy. On the other hand, a study of synchronized digital stethoscope acquisition and fiberoptic endoscopic evaluation of swallowing [11] suggested that there was no consistent association of CA signals with FEES images. In other words, as the counterevidence to the cardiac analogy hypothesis, there was no clear one-to-one correspondence between swallowing movements and audible sounds.

### 1.3. The Acquisition of Swallowing Mechano-Acoustic Signals: The Know-How

Based on the assumptions of the nature of swallowing mechano-acoustic signals discussed above, the site(s) of acquisition adopted for single-point SA and CA have mainly followed the work of [2], subsequently supported by [3]. Takahashi and colleagues [2] compared the peak intensity and frequency of signals sequentially acquired on 24 sites around the neck region and concluded three promising sites for acquiring mechano-acoustic signals of swallowing: (i) the space between cricoid cartilage and immediately superior to the jugular notch along the midline, (ii) the center of cricoid cartilage, and (iii) the lateral border of trachea immediately inferior to the cricoid cartilage. Subsequently, as a revisit, a study using microphones and accelerometers [3] recommended that (ii) would be the most optimal site, followed by (i) and (iii). Since then, these sites have been extensively used in both CA and SA, e.g., site (ii) for the accelerometer of HRCA in [9,12], dual-axis accelerometer in [13], microphone in [14], and site (iii) for microphone of HRCA in [9,12], condenser microphone in [15], and digital stethoscope in [16]. It should be noted that, based on the assumed nature of symmetry and reproducibility [1], the conclusions of optimal sites were drawn from recordings taken sequentially one after another. More recently, Pan and colleagues [17] revisited the optimal site problem using a 5 × 4 simultaneous microphone array and concluded a different site of acquisition to be the inferior border of the mandible immediately superior to the strap muscle. Despite coming to an alternative conclusion of the optimal site of acquisition, their study still adhered to the assumed nature of symmetry in swallowing mechano-acoustic signals.

### 1.4. The Current Study

By now, it seems that the large body of mechano-acoustic studies have rested upon the assumption that swallowing signals are grossly symmetrical and reproducible regardless of the sensor(s) used, despite a lack of a clear consensus as to the exact cause(s) of swallowing sounds and vibrations. The current study, therefore, aimed to revisit the nature of swallowing mechano-acoustic signals and hence their possible underlying physiological process in the absence of such assumptions. Specifically, as an extended revisit of the findings in [1], the study examined the consistency (reproducibility) and coherence (symmetry) of swallowing mechano-acoustic signals, and thus the possible underlying physiological process, using a conformal swallowing accelerometry (CSA) system. Conceivably, this is the first study to adopt a multi-channel paradigm that permits the simultaneous acquisition of mechano-acoustic signals of swallowing, leading to a refined understanding of the nature of the underlying physiological process, and hence informing future design of biomedical sensors for swallowing screening and monitoring.

## 2. Materials and Methods

### 2.1. Participants

The study was conducted in accordance with the Declaration of Helsinki and approved by the Institutional Review Board of the Hong Kong Polytechnic University (Ref. No. HSEARS20220816002). Thirteen healthy young adults, including eight females (mean age = 24.38 ± 0.92) and five males (mean age: 24 ± 3.74), were recruited by convenience sampling in the Hong Kong Polytechnic University from September to November 2022. The participants reported neither medical diagnoses nor known history associated with dysphagia, and passed the 3-ounce water screening test [18].

### 2.2. System Architecture and Sensor Calibration

There were three major components in the CSA system, namely mechano-acoustic sensors, the respective micro-controller boards, and a MATLAB (R2023a, The MathWorks, Inc., Natick, MA, USA) post-processing program. The mechano-acoustic sensors were eight tri-axial accelerometers (ADXL 335, Analog Devices, Norwood, MA, USA). The ADXL 335 is a tri-axial accelerometer known for its compact dimensions (4 mm × 4 mm × 1.45 mm), lightweight design (1.27 g), extremely low background noise level (maximum 0.00294 ms^2^/√Hz rms), high sensitivity (0.031 ms^2^/mV), and minimal power consumption of 320 µA. It measured the surface vibratory acceleration normal to the neck surface with a minimum full-scale range of ±3 g, operating under a single-supply voltage that ranged from 1.8 V to 3.6 V. Each accelerometer was conditioned and controlled using a compact Arduino Nano microcontroller board, which was based on an 8-bit AVR RISC architecture (Microchip Technology model ATmega328) and featured 32 KB of in-system programmable flash memory with read-while-write capabilities. The Arduino Nano was programmed through the Arduino Software integrated-development environment to acquire skin-surface vibration outputs from the accelerometers over a duration of 10 s. The controller was configured to read voltage inputs ranging from 0 V to 5 V and convert each input into a representative numerical value that spans from 0 to 1023. The sensitivity of each accelerometer, expressed in (m/s^2^)/V or g/V, was determined by utilizing a precision vibration calibration exciter (Brüel & Kjær type 4294). This calibration involved comparing the output of the accelerometer to the measurement results obtained from a high-fidelity miniature piezoelectric accelerometer (Brüel & Kjær type 4397) that was subjected to the same excitation conditions.

### 2.3. Raw Signal Acquisition and Processing Pipeline

Figure 1 presents a concise overview of the acquisition pipeline of swallowing mechano-acoustic signals. The process was initiated by a push switch and automatically halted after the designated 10-s duration. Before sending the acquired swallowing vibration data for further signal analysis, it was processed using custom codes developed for execution in MATLAB (R2023a, The MathWorks, Inc., Natick, MA, USA) software for two purposes. The first one involved removing occasional erroneous spikes, likely caused by temporary partial detachment of the sensor from the skin, while preserving the essential features of physiological swallowing in the captured acceleration signal (Figure 2). The removal was achieved with non-linear median filtering technique available in MATLAB (R2023a, The MathWorks, Inc., Natick, MA, USA) which was designed for eliminating impulsive noise from one-dimensional time-series data. The second involved a procedure that converted the captured acceleration time trace, represented by sensor output values, into temporal fluctuations expressed in physical units of m/s^2^ or g. This approach not only enhances the control of potential measurement errors but also establishes a consistent physical framework that facilitates improved data interpretation. Such a framework allows for easier comparison and understanding of the impact of various swallowing parameters and trial settings.

### 2.4. Data Acquisition

The CSA system acquired mechano-acoustic signals simultaneously at 8 sites along the neck and one side of the strap muscle, with one accelerometer on each site (see Figure 3 for illustration). To investigate the coherence (symmetry) of the signals and the associated motions, 6 of the 8 sites were paired on both sides of the neck, namely (i) the superior horn of the thyroid cartilage, (ii) the lateral border of thyroid lamina, and (iii) the lateral border of the cricoid cartilage. These sites were chosen to cover the whole laryngeal complex (e.g., laryngeal elevation discussed in [2]) with well-defined anatomical landmarks while being obstruction-free at the midline. The placement of accelerometers on these 6 sites altogether formed a vertically aligned bilateral array on the neck. The remaining single-sensor sites were as follows: (iv) the space immediately inferior to the cricoid at the level of the first or second tracheal ring, and (v) the left/right clavicle attachment of the strap muscle. Site (iv) had been reported to offer optimal acoustic information (e.g., signal-to-noise ratio) about swallowing activity [2,3], and site (v) was a control point. The anatomical landmarks were determined by palpation. To ensure optimal transmission of vibration signals, the accelerometers were placed at either the inter-structure junctions or the muscular part of the respective structure [5]. For the bilateral vertically aligned array, i.e., sites (i) to (iii), the horizontal and vertical intervals between each accelerometer on sites (i) and (iii) were respectively standardized at approximately 20 mm and 35 mm. For simplicity, we shall now refer to Channels 1, 2, and 3 as the sensors, respectively, at the superior thyroid horn, thyroid lamina, and lateral cricoid positions on the left, and accordingly, Channels 5, 6, and 7 to the symmetrical sensors on the right. Channel 4 refers to the inferior cricoid sensor, and Channel 8 refers to the control point at the strap muscle.

The following steps were sequentially taken to ensure tight contact of the accelerometer with skin surface and avoid air gaps: (i) neck surface cleansing with alcohol wipes, (ii) initial attachment of accelerometers to the skin surface of the abovementioned sites with double-sided tapes, and (iii) further cover and affixation with surgical tape (3M^TM^ Micropore^TM^ Surgical Tape 1530-1, Denver, CO, USA).

The sampling rate was fixed at 3600 Hz. The determination of the sampling rate was decided with reference to the reported peak frequency of swallowing tasks in previous studies utilizing dual- or tri-axial accelerometers [10,19,20], coupled with oversampling to prevent possible aliasing effect.

During signal acquisition, the participants performed swallowing of thin water boluses of 5 mL and 10 mL respectively, 3 times in separate repeated trials while sitting comfortably in an upright position. For each trial, the participants were instructed to make a hum (/m/) before and after the task. The humming served as brackets for extracting task-relevant signal segments during preprocessing and analysis.

### 2.5. Data Extraction

Given that the current study first targeted revisiting the characteristics of swallowing mechano-acoustic signals, only signals from sites (i) and (iii) were extracted and analyzed to examine their consistency (reproducibility) and coherence (symmetry). While site (iv) has been reported as one of the optimal sites in previous studies, the extraction and analysis were reserved for subsequent studies should asymmetry be proven in the current study. In addition, only vibration signals orthogonal to the skin surface (i.e., anterior–posterior) were extracted and analyzed.

#### 2.5.1. Time Domain Analysis

Peak intensity (PI) and peak intensity time (PIT) are common dynamic measures for evaluating the magnitude and time of occurrence in SA studies [21]. Signal intensity was usually opted for as a measure since it represents the magnitude of acceleration (m/s^2^) of a motion at a certain point and direction. Provided that the magnitude of force is proportional to acceleration from a Newtonian-mechanics perspective, the PIs of an accelerometer signal then represent the maximal force of skin-surface vibrations induced from swallowing-related movements at the point of data acquisition. For this study, PI was defined as the maximum intensity in the time domain and PIT as the timestamp at which PI took place. It must be noted that the time-averaged value of raw temporal acceleration signals for each swallowing event was removed to enhance the quality of signal dynamic features for calculating PI and PIT. The subplot at the top of Figure 4 illustrates an annotated sample accelerometer signal.

It should be noted that extraction and analysis in this study only focused on the anterior–posterior (A-P) acceleration due to the vibration/movement orthogonal to the skin surface. The mechano-acoustic features of the A-P signals have been shown to be correlated with swallowing movements such as hyolaryngeal excursion (e.g., [22]). For superior–inferior (S-I) acceleration, it was expected that the vertically aligned array could capture the respective dynamics, although previous studies also showed its significant correlation with swallowing movements (e.g., [19]).

#### 2.5.2. Time-Frequency Analysis

Previous studies have also used peak frequency (PF) to characterize accelerometer signals (e.g., [8,23,24,25]). PF has been defined in two different ways. The first defines PF as the frequency with the highest power identified at the PIT in the spectrogram (e.g., [24]). The second defines PF as the frequency with maximal spectral power identified by time-frequency analysis (e.g., [8]). In this study, the second definition was adopted. Under this definition, PF essentially represents the phase of an independent cycle of motions with maximum power at a specific time point in a signal. In turn, it offers information about a particular component of the physical process that gives rise to the waveforms in the signal. For the current study, continuous wavelet transform (CWT) with the complex Morlet wavelet was used to decompose the signals. CWT was chosen over short-time Fourier transform (STFT) adopted in previous studies (e.g., [25]) for its adaptive time window, thus conferring better time and frequency localization for non-stationarity and higher resistance to numerical noise. Peak frequency time (PTF), defined as the respective time of PF, was also extracted. The implementation of time-frequency analysis was conducted in Python 3.9.9 using the PyWavelets library [26]. A CWT sample is shown in the scaleogram at the bottom of Figure 4.

### 2.6. Hypothesis Testing and Statistical Analysis

As a revisit to [1], the consistency (reproducibility) and coherence (symmetry) of the swallowing mechano-acoustic signals and their possible physiological source were investigated using statistical measures. The signals and their source would be considered as more consistent/reproducible if the analysis indicated insignificant statistical difference and high inter-trial agreement between PI and PF across trials and volumes. Similarly, the signals and their source would be considered as more coherent/symmetrical if at least contra-lateral channel agreement was indicated for PI and PF.

The data analysis procedures were as follows. First, three-way (3 trials × 2 volumes × 6 channels) ANOVAs with repeated measures (or a non-parametric equivalent) were conducted to examine statistical differences in PI and PF across/within trials and channels. This linear mixed model aimed to investigate any main effect of trial (inter-trial–intra-channel difference), channel (intra-trial inter-channel difference), bolus volumes, and their interaction. Upon examining statistical differences, Coefficients of Variation (CVs) were calculated as a descriptive measure of reproducibility/consistency. CV was defined as the ratio of standard deviation to the respective mean. After that, two-way mixed-effect Intraclass Correlations (ICCs) and two-way random-effect ICCs were used. Inter-trial intra-channel (for reproducibility/consistency) and intra-trial inter-channel (for coherence) agreement were evaluated, respectively, using the PIs and PFs. The ICC agreement analysis was further complemented by Bland–Altman plots to graphically examine the 95% Confidence Interval (CI) Limits of Agreement (LOAs). Furthermore, for inter-channel comparison, the temporal coherence (i.e., the extent to which PI and PF across channels took place within a specific timeframe) was also evaluated by the correlation coefficients and ICCs of intra-trial inter-channel PIT and PFT.

## 3. Results

### 3.1. Descriptive and Inferential Statistics

A total of 468 accelerometer signals (6 channels × 3 repeated trials × 2 bolus volumes × 13 participants) were acquired and analyzed. Figure 5 and Table 1 summarize the descriptive statistics of PI and PF. As shown in Table 1, a majority of the PI and PF values did not follow a normal distribution statistically, and hence, non-parametric tests were subsequently used. In general, there was a wide range of values with some outliers for both variables, noticeably in larger volumes (see Figure 5). The values of PI spanned from as low as 0.52 (m/s^2^) to as high as 52.95 (m/s^2^), and PF from 3.76 Hz up to 602.13 Hz.

For inferential statistics, a three-way (3 trials × 2 volumes × 6 channels) Aligned Rank Transformation (ART) ANOVA with repeated measures [27] revealed significant statistical differences in PI among repeated trials with a small effect size (*F*(2420) = 3.170, *p* < 0.05, *η_p_*^2^ = 0.015) and among channels with a moderate effect size (*F*(5420) = 5.902, *p* < 0.001, *η_p_*^2^ = 0.066). Tukey’s HSB post hoc comparison indicated marginal statistical difference between Trial 3 and two other trials (*p* = 0.089), and significant statistical difference between Channel 7 with all other channels (all *p* < 0.01) but its contralateral counterpart, i.e., Channel 3. No interaction effect was indicated. Similarly, a three-way (3 trials × 2 volumes × 6 channels) ART ANOVA with repeated measures also suggested significant statistical difference in PF across trials (*F*(2420) = 4.163, *p* < 0.05, *η_p_*^2^ = 0.019) and channels (*F*(5420) = 2.281, *p* < 0.05, *η_p_*^2^ = 0.026) with a small effect size. Furthermore, a statistically significant interaction effect of volume and trial with a small effect size (*F*(2420) = 5.154, *p* < 0.01, *η_p_*^2^ = 0.024) was indicated. Post hoc comparison merely indicated significant difference between Trial 2 and 3 (*p* < 0.05), with no significant differences in PF among channels (all *p* > 0.05). An actual interaction effect of volume with repeated trials only existed between one of the 10 mL and 5 mL trials (*p* < 0.05).

### 3.2. Consistency (Reproducibility)

Figure 6 illustrates the distributions of CVs of PI and PF of each channel across trials. Overall, a majority of the CVs were higher than 20% in the repeated trials. Compared to PI, higher dispersion was noted in PF, whose CVs were 50% or above. This 369 held true even when different volumes were considered. In addition, both domains presented a few samples of extreme cases where CV was 100% or above, particularly in the superior thyroid channels (Channels 1 and 5) in PF, suggesting instances of high dispersion of observed values in repeated trials. As indicated by the ICCs in Table 2, there was generally poor agreement of both PI and PF among repeated trials, except PI at Channel 1. A more detailed breakdown of ICCs across volumes (see Appendix A) showed that PIs of 10 mL bolus tended to exhibit higher agreement in Channels 1, 3, 5, and 6 than those of 5 mL bolus in Channels 2 and 7. Comparatively, apparently higher agreements were only shown in PFs of 10 mL bolus in Channels 1 and 3.

On the other hand, as the channel-wise trial-by-trial mean-difference plots (see Figure 7) illustrate, a large majority of such poor agreements laid within the 95% CI LOAs. Only a few points of difference were beyond the LOA in each channel. No obvious difference was noticed for different volumes graphically. Notably, the PI and PF of Channels 3 and 7 (lateral cricoid), except for one of the trials on the right (Trial 3), had the narrowest inter-trial LOA of less than 10 m/s^2^ and around 100 Hz or less, respectively.

### 3.3. Coherence (Symmetry)

Similar to findings in reproducibility/consistency, there was poor agreement with the expected variance in intra-trial inter-channel PIs and PFs. As the ICCs indicated, both PI and PF showed poor agreement among the channels, except moderate agreement among the PIs of superior thyroid (Channels 1 and 5) and lateral cricoid (Channels 3 and 7) (see Table 3). With respect to differences in agreement across volumes (see Appendix A), PIs in thyroid lamina (Channels 2 and 6) exhibited higher agreement in 5 mL than in 10 mL bolus. This was also apparent in ipsilateral agreement. On the other hand, such volume difference was not observed in the frequency domain.

Like inter-trial differences, the mean-difference plots showed that a majority of such poor agreement again fell, respectively, into the 95% LOAs of up to absolute values of 12 m/s^2^ and 240 Hz for PI and PF. For PI, the mean absolute differences were less than 2 m/s^2^. For PF, the mean differences varied between 1 and 27 Hz, indicating inter-channel systematic bias in the frequency domain.

Specifically, the inter-channel differences in the frequency domains varied at different heights of the neck. At the superior thyroid level (Channels 1 and 5), the mean differences between PFs on the left and all other channels were approximately 10 Hz or higher, with wide LOAs of almost 200 Hz or higher, while there were smaller differences and narrower LOAs on the right. At the thyroid lamina level (Channels 2 and 6), the mean differences between PFs of Channel 2 and those of all others but the right counterpart were positive. However, at the lateral cricoid level (Channels 3 and 7), the PFs only differed by an average of 1.5 Hz, with narrow LOAs of 100 Hz or less, while the mean differences with other channels were higher. In addition, a U-shaped scattering pattern was noticed in both the inter- and intra-trial mean-difference plots of PI and PF, with those of PF most pronounced (See Figure 8 and Figure 9). Regarding volume difference, no particular patterns were observed graphically.

Temporal Coherence: The scatterplot visualizations of the pairwise agreements of PI and PF and their respective timings across volumes are shown in Figure 10. As illustrated by the scatter plots and Table 4, it was apparent that, on the one hand, both PI and PF showed poor to moderate agreement and weak correlations (except superior thyroid channels), but their respective peak times showed, on the other hand, excellent agreement and strong correlations. This remained true even for different volumes (see Appendix A) as well as inter-domain and inter-channel gross comparisons, with excellent agreement suggested by a two-way random effect ICC of 0.966 with a 95% CI from 0.95 to 0.98 (F(77,847) = 338.151, *p* < 0.0001). The only exception was the moderate agreement of 5 mL bolus peak intensities of Channels 2 and 6.

## 4. Discussion

This study set out to revisit the longstanding proposal of consistency/reproducibility and coherence/symmetry of mechano-acoustic signals from swallowing accelerometry and their underlying physiological process(es). Overall, the current study has led to two significant implications regarding the characteristics of the physiological process(es) that underpin swallowing mechano-acoustic signals, and hence their acquisition techniques. Firstly, similar to previous findings, the current study provided further evidence for intra-personal variability in the magnitude and dynamics of swallowing signals with additional effect from increased bolus volume, and hence for the necessity of repeated trial measurements with different bolus sizes. Secondly, multiple incoherent/asymmetric movements or physical processes, as indicated by poor intra-trial inter-channel PF agreement, might have taken place at near-concurrent time during swallows. Such asymmetry also seemed to be generally pervasive across bolus volumes. This was in stark contrast with previous findings in [1] while echoing Heinz’s [29] theoretical proposal. Although the current study merely drew on a small and homogenous sample from healthy young adults, the new preliminary evidence of the asymmetry of swallowing signals lends support to the use of multi-point rather than single-point accelerometry to characterize swallowing dynamics. To the best of the authors’ knowledge, [1] is the only study to date to investigate the symmetry and reproducibility of swallowing acoustic signals using concurrent and symmetrical submental and laryngeal accelerometry measurements. Therefore, the findings and implications will be discussed with respect to their work.

Intra-personal variability of swallowing signals, suggested previously by [1], was further evidenced by statistical results as well as inter-trial analyses of the magnitude (PI) and dynamics (PF) in the current study. Statistical results in inter-trial differences (for both PI and PF) and the interaction with volume effect (for PF) implied expected variability within individuals and were grossly consistent with other variability measures. It was probable that a combination of repeated trials and different volumes would suffice to capture a holistic mechano-acoustic profile of swallowing physiology. With respect to such differences and variability, PI and PF were found to vary within expected (95% CI) limits with very few outliers, in spite of poor to moderate inter-trial agreement. Regarding measurement errors of PF in repeated trials, the findings were similar to [1] with higher variability, i.e., most CVs were higher than 20%. This might be attributed to the difference in time-frequency analysis, where STFT with 100 msec Hamming windows were adopted, whereas the current study employed CWT whose adaptive window sizes might offer more precise time and frequency localizations. Despite such methodological differences, the current findings supported the previous proposal that repeated trial measurements are necessary for capturing any intra-personal variability in swallowing movements ([1]).

It should be highlighted that little systematic bias (mean difference) was noted in the time domain (less than 2 m/s^2^), but was evident in the frequency domain (varying between 2 and 47 Hz). This observation was coupled with U-shaped or moon-shaped mean-difference plots in inter-trial PFs, suggesting a systematic variance or error that differences and fluctuations widen and intensify as frequency increases (see Figure 9). This pattern, however, was not explicit in inter-trial PIs. This might imply that, within healthy individuals, the magnitude of the intra-laryngeal traction could remain stable or constant while its dynamics could exhibit a high degree of flexibility or variability across trials. The force exerted, represented in acceleration, by swallowing movements might be partially independent of their motion patterns. Conceptually, given the cardiac analogy hypothesis [5], this could also be due to the alterations in the configuration of the oropharynx during swallowing, e.g., squeezing of the pharyngeal walls. This could be true if we consider the findings that the inter-trial variation and wider LOAs in the PFs at the superior thyroid and thyroid lamina were higher than that at the lateral cricoid channels.

This inter-trial variation and partial independence of PI and PF were also evident across different channels in the same trial, suggesting that the force and dynamics of the underlying physiological process(es) tended to be incoherent/asymmetric. The interesting results of the statistical difference between the PI of Channel 7 and all channels and Channel 3 on its contralateral side suggested that the PI of the lateral cricoid varied within the same distribution/range of values. In a similar vein, their PIs were most agreed in inter-channel comparison (ICC = 0.612). However, except for the agreement between superior thyroid channels (i.e., Channels 1 and 5), such results did not apply to other measurements (i.e., PF) and contralateral pairs, suggesting the likelihood of signal peak asymmetry across different bolus sizes. These findings were in stark contrast with the previous proposal regarding the symmetry of mechano-acoustic signals of swallowing. Takahashi and colleagues [1] investigated the nature of swallowing signals using simultaneous bilateral accelerometry on the lateral side of the cricoid cartilage. Based on the descriptive statistics of CV of the first and second PFs, and the duration of swallows, they concluded that swallowing signals, although varying across trials, were grossly symmetrical at the lateral cricoid level, and thus a repeated-measure single-point approach would suffice to characterize swallowing motions using accelerometry. The current findings, on the other hand, suggest the alternative approach of repeated-measure multiple-point acquisition.

Findings of temporal coherence between PI and PF further reinforce such an alternative proposal of multi-channel acquisition of mechano-acoustic swallowing signals. As clearly illustrated in the normalized scatterplots (Figure 10) and the intra-trial agreement and correlation metrics, both PIT and PFT tended to take place at roughly the same time point, whereas, as discussed above, the forces and accelerations induced by swallowing movements were comparatively less symmetric/coherent. Together with the partial independence of signal magnitude and dynamics, which could vary up to an absolute difference of 200 Hz, as well as the systematic bias identified in the U-shaped Bland–Altman plots, it was likely that multiple concurrent movements were taking place across the sites of acquisition with their own dynamics, hence coherence in time but incoherence in magnitude and dynamics. Leslie and colleagues [11] in fact concluded from their concurrent FEES and digitalized stethoscope auscultation study that there might not exist a one-to-one signal–movement correspondence in the context of cervical auscultation. The current findings of magnitude and dynamic incoherence/asymmetry with temporal coherence further support this conclusion.

Furthermore, ipsilateral dynamics were also shown to be non-uniform (or non-linear), as indicated by the Bland–Altman plots of intra-trial PFs. Despite the absence of superior–inferior signals, this could still imply that signals on the same side did not propagate from one point to another in a certain direction. In turn, the directionality underlying the process could be uncertain, or there could actually exist multiple processes/movements on the same side. This further suggests that simultaneous and symmetrical placements of acoustic sensors at certain levels of the neck (e.g., at the lateral cricoid in [1]) might still not suffice to capture the laryngeal dynamics during swallowing. In other words, the results from CSA have also lead us to rethink the necessity of pursuing one single optimal site for mechano-acoustic signal acquisition (e.g., [2,3]); it may be necessary instead to find an optimal CSA configuration/alignment that is capable of capturing the complexity and intricacy of swallowing dynamics, which could conceptually be an ensemble of multi-point concurrent kinematic processes [28].

## 5. Limitations and Future Research

A few limitations should be noted carefully in the present study, particularly in terms of methodology. Firstly, as a pilot study, only a small group of healthy young individuals in a specific population and age group was recruited. This could be prone to sampling bias. Future research shall address this issue with a larger sample with a more diverse population. The inclusion of healthy older adults and dysphagic individuals would also further validate the proposal in the current study. Secondly, unlike previous studies like [11], only SA was used as a standalone measure, and hence the origin of the signals remained unclear. Synchronization with other imaging techniques would further unveil and verify the swallowing event(s) associated with the peak intensity and frequency of the signals. Thirdly, the PIs and PFs from the inferior cricoid, which has been deemed as one of the optimal sites (e.g., [2,3]) and hence commonly adopted in other studies (e.g., [9,12,15,16]), were not extracted and analyzed. Given that this study was meant to be a revisit to a previous proposal concerning reproducibility and symmetry, the analysis of inferior cricoid signals in combination with other sites shall be pursed in a follow-up study.

## 6. Conclusions

Swallowing accelerometry has been adopted as a non-invasive technique in research to detect, characterize, and evaluate movements of deglutition in the past decades. This technique has been implemented using a (unilateral) single-point method, based upon the assumption of signal symmetry [1] and previous findings of the optimal site of acquisition [2,3]. As a revisit to the symmetry and reproducibility of swallowing acoustic signals, this study agreed with the previous findings of intra-personal variability. However, it has demonstrated preliminary evidence for an alternative proposal of bilateral multi-point acquisition with the use of conformal swallowing accelerometry. The temporal coherence coupled with non-uniform and incoherent dynamics of mechano-acoustic swallowing signals has led us to rethink the current methodology of single-point and optimal-site acquisition for swallowing signals. As an alternative pursuit, a multi-channel conformal accelerometry could be a viable and promising approach for future sensor designs and implementation of swallowing mechano-acoustic signal acquisition for dysphagia screening and monitoring.

## 7. Patents

The design and implementation of CSA as non-invasive instrumental swallowing evaluation described in this study is related to a U.S. Provisional Patent Application filed by the authors. The details of the patent are as follows:

Leung Chi Kin Randolph, Kwong Yee Lan Elaine, Lam Yiu Shun Wilson, Rai Sanjaya, Lui Kwan Leo. “Multi-channel Conformal Swallowing Accelerometry Technology For Detection And Diagnosis Of Dysphagia” (Application No.: 63/812,713), 23 May 2025.

## Figures and Tables

**Figure 1 sensors-25-07396-f001:**
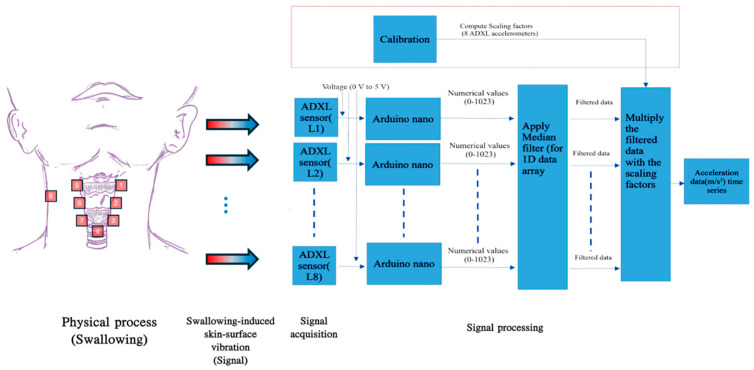
A schematic view of the swallowing acceleration measurement pipeline.

**Figure 2 sensors-25-07396-f002:**
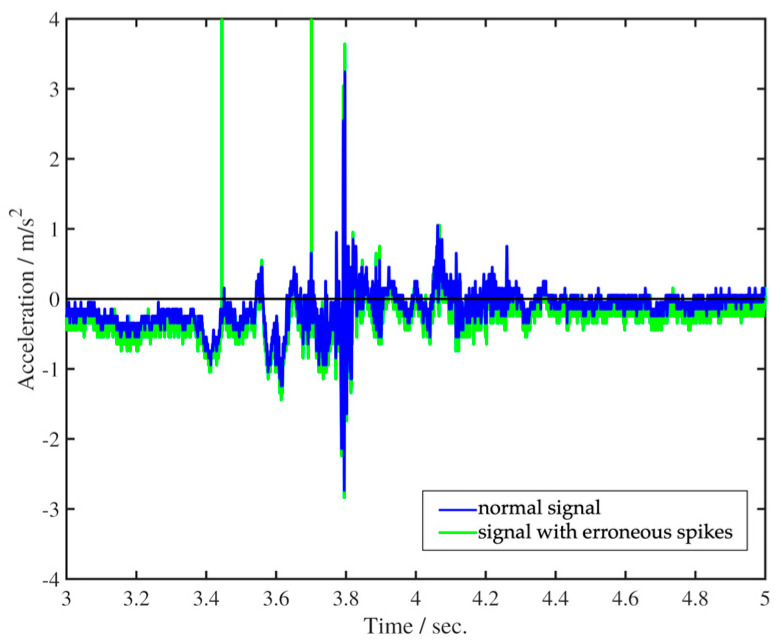
Removal of erroneous spikes from vibration acceleration time trace.

**Figure 3 sensors-25-07396-f003:**
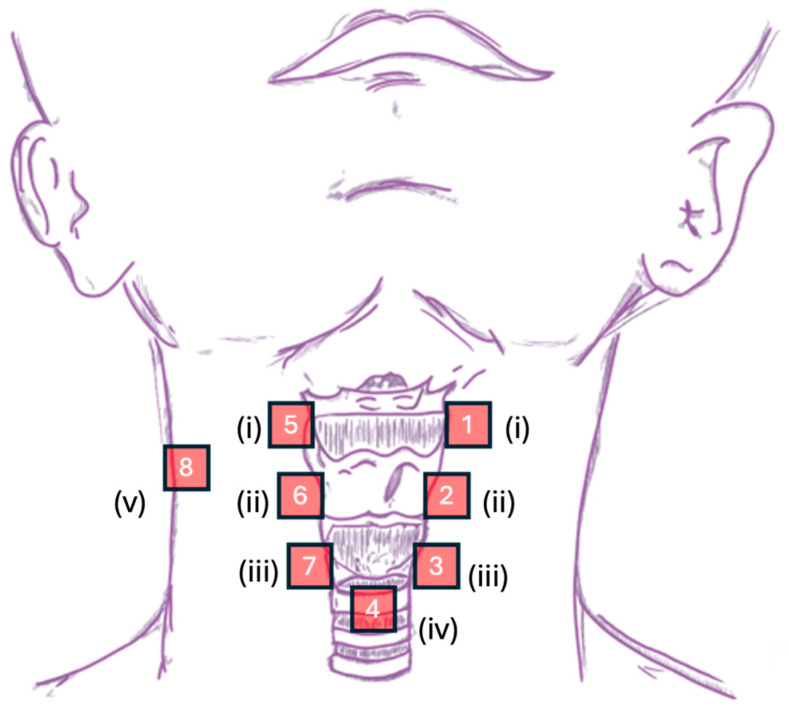
Graphical representation of sensor placement in the CSA system— roman numerals: (i) the superior horn of the thyroid cartilage, (ii) the lateral border of thyroid lamina, (iii) the lateral border of the cricoid cartilage, (iv) the space immediately inferior to the cricoid at the level of first or second tracheal ring, and (v) the left/right clavicle attachment of the strap muscle.

**Figure 4 sensors-25-07396-f004:**
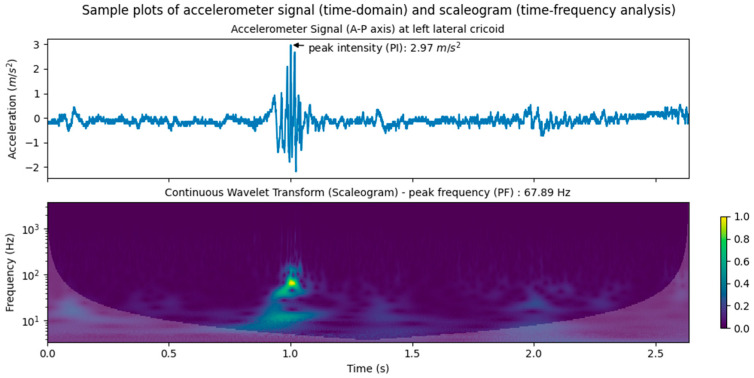
Sample plots of accelerometer signal with PI annotated (**top**) and CWT scaleogram (**bottom**) at left lateral cricoid.

**Figure 5 sensors-25-07396-f005:**
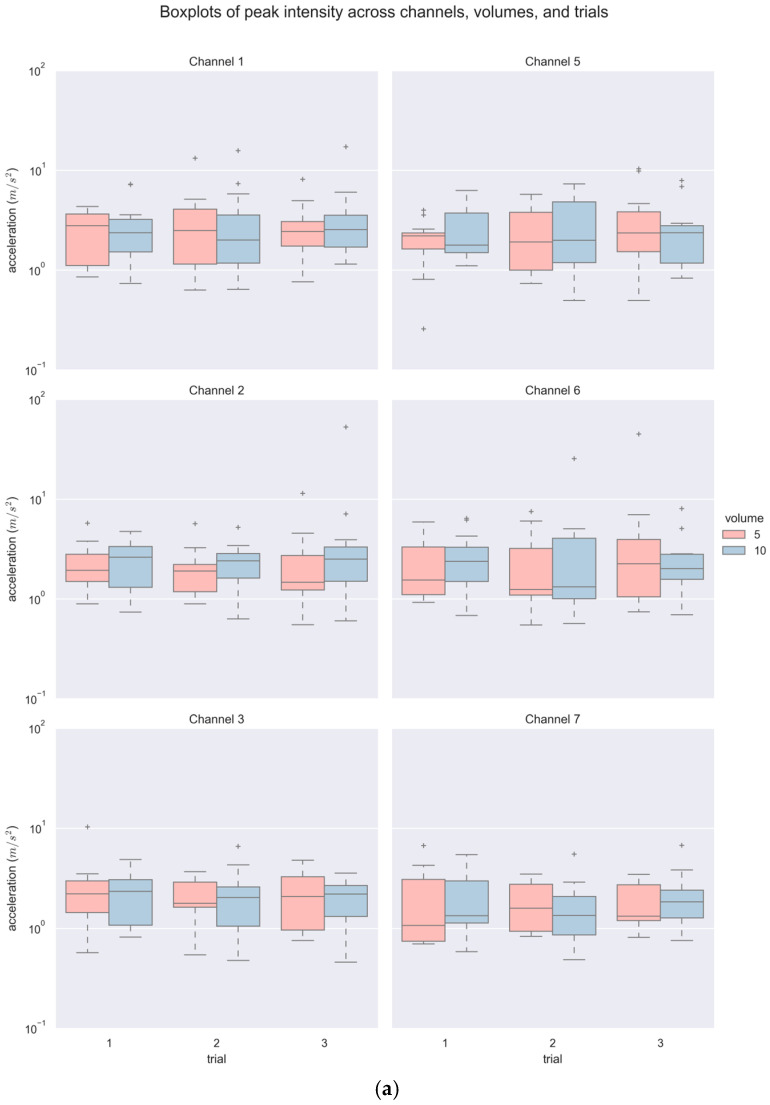
Boxplots of (**a**) PI and (**b**) PF across channels, volumes, and trials.

**Figure 6 sensors-25-07396-f006:**
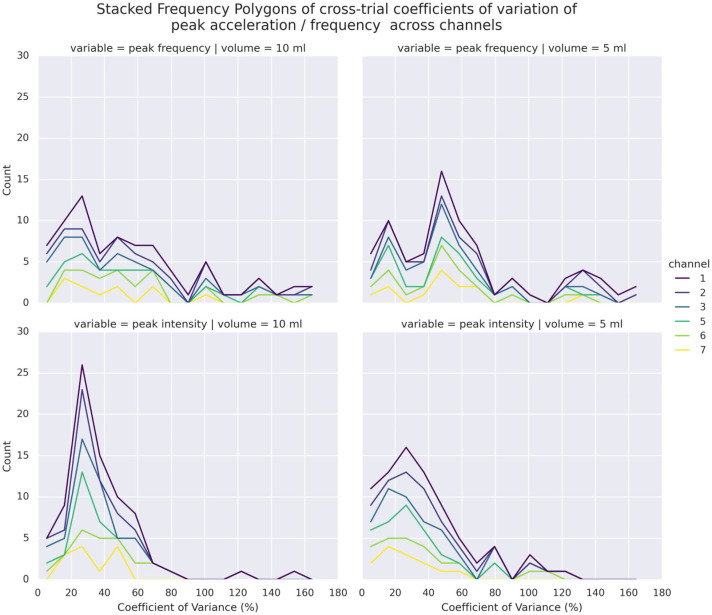
Stacked Frequency Polygons (Histograms) of CVs in PF (**top**) and PI (**bottom**) of 5 mL (**right**) and 10 mL (**left**) bolus across trials and channels.

**Figure 7 sensors-25-07396-f007:**
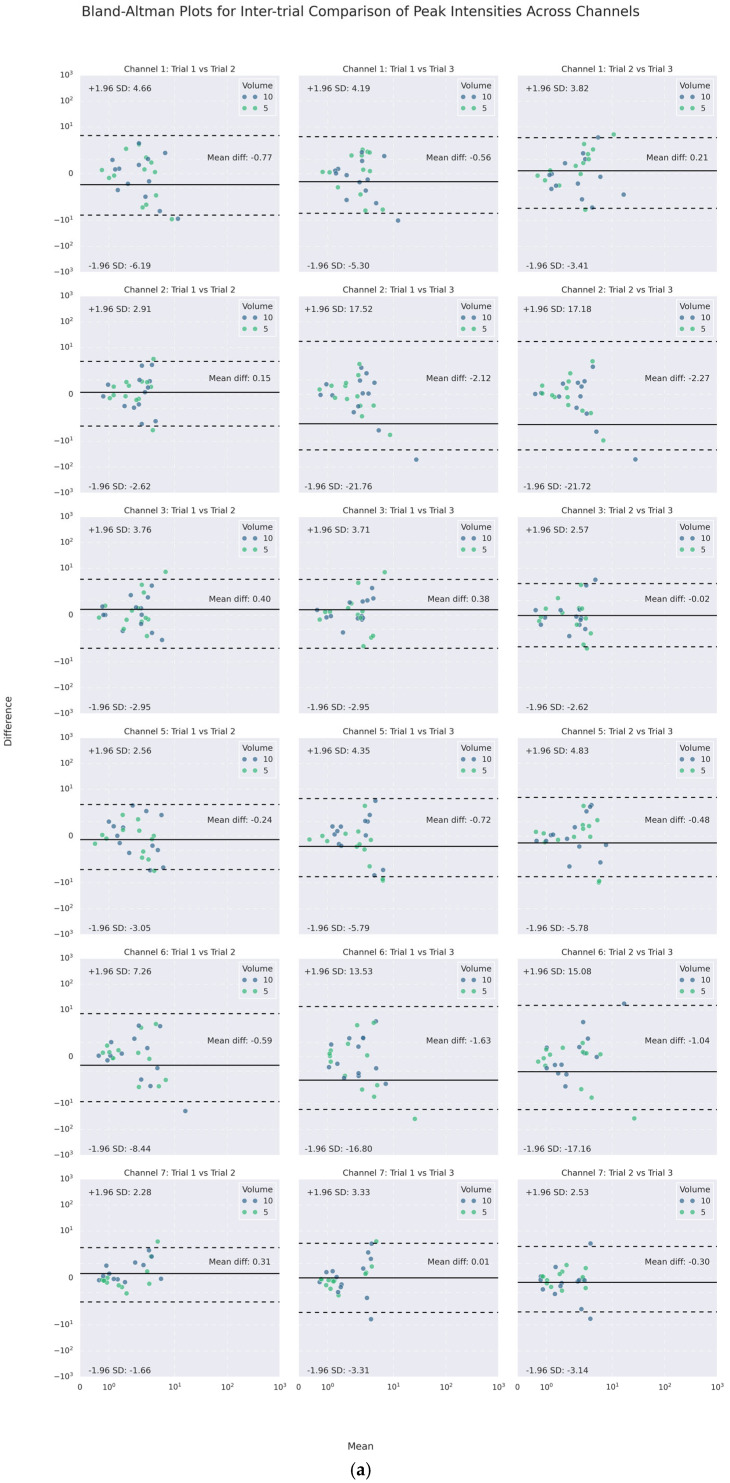
Channel-wise Bland–Altman mean-difference plots of inter-trial (**a**) peak intensities and (**b**) peak frequencies.

**Figure 8 sensors-25-07396-f008:**
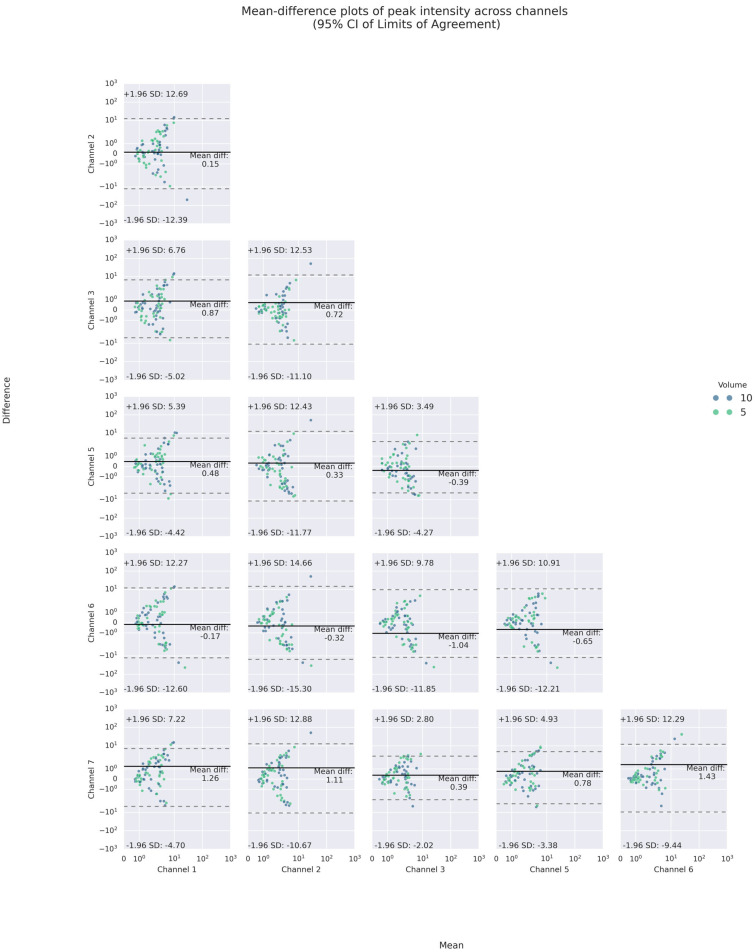
Bland–Altman mean-difference plots of pairwise comparisons of PI.

**Figure 9 sensors-25-07396-f009:**
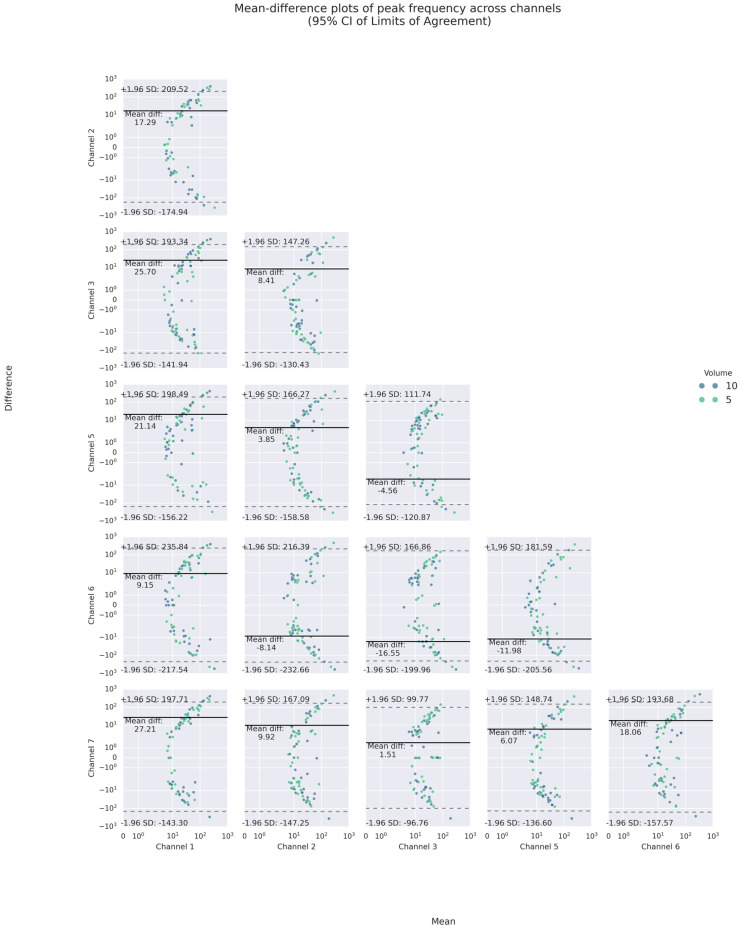
Bland–Altman mean-difference plots of pairwise comparisons of PF.

**Figure 10 sensors-25-07396-f010:**
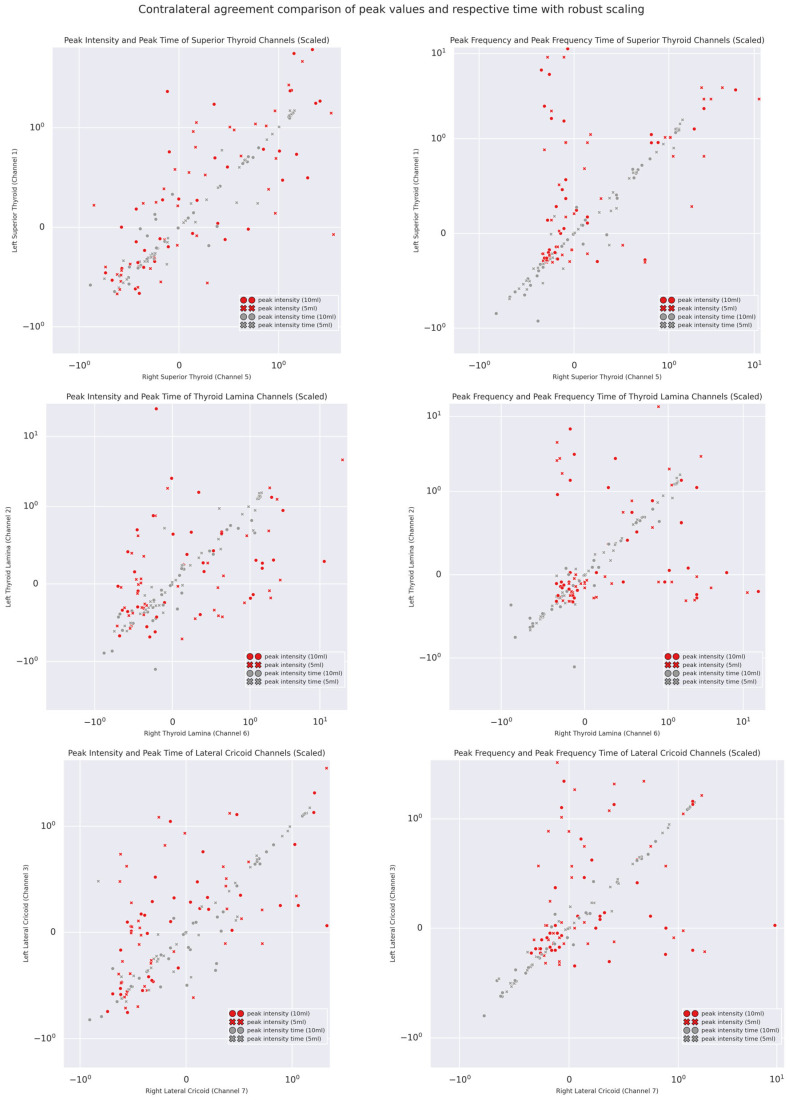
Normalized scatterplots of contra-lateral intra-trial peak intensity, peak frequency, and their respective times.

**Table 1 sensors-25-07396-t001:** Descriptive statistics of PI and PF of each channel across volumes and trials.

			Trial 1 N = 13		Trial 2 N = 13		Trial 3 N = 13	
Variable	Channel	Volume (mL)	Median (IQR)	Shapiro–Wilk Statistic	Median (IQR)	Shapiro–Wilk Statistic	Median (IQR)	Shapiro–Wilk Statistic
Peak intensity (m/s^2^)	1	5	2.792 (2.534)	0.887 ^n.s.^	2.494 (2.933)	0.724 ***	2.442 (1.331)	0.817 *
		10	2.376 (1.715)	0.806 **	2.004 (2.39)	0.705 ***	2.556 (1.848)	0.624 ***
	2	5	1.941 (1.310)	0.844 *	1.913 (1.028)	0.784 **	1.475 (1.5)	0.641 ***
		10	2.634 (2.058)	0.927 ^n.s.^	2.417 (1.226)	0.934 ^n.s.^	2.517 (1.806)	0.409 ***
	3	5	2.221 (1.537)	0.668 ***	1.786 (1.26)	0.960 ^n.s.^	2.087 (2.304)	0.890 ^n.s.^
		10	2.354 (1.987)	0.908 ^n.s.^	2.041 (1.527)	0.843 *	2.209 (1.356)	0.950 ^n.s.^
	5	5	2.209 (.730)	0.959 ^n.s.^	1.921 (2.791)	0.889 ^n.s.^	2.368 (2.309)	0.774 **
		10	1.784 (2.236)	0.875 ^n.s.^	1.999 (3.659)	0.882 n^.s.^	2.375 (1.618)	0.757 **
	6	5	1.555 (2.213)	0.815 *	1.251 (2.118)	0.800 **	2.258 (2.911)	0.449 ***
		10	2.388 (1.802)	0.877 ^n.s.^	1.331 (3.07)	0.514 ***	2.022 (1.234)	0.771 **
	7	5	1.07 (2.325)	0.714 ***	1.593 (1.817)	0.860 *	1.328 (1.518)	0.867 *
		10	1.34 (1.847)	0.836 *	1.348 (1.221)	0.773 **	1.849 (1.136)	0.807 **
Peak Frequency (Hz)	1	5	11.835 (37.786)	0.484 ***	24.07 (46.384)	0.639 ***	48.839 (76.821)	0.844 *
		10	37.246 (40.449)	0.636 ***	18.316 (49.015)	0.682 ***	21.55 (47.268)	0.742 **
	2	5	12.497 (6.722)	0.660 ***	13.943 (49.608)	0.785 **	15.554 (100.153)	0.600 ***
		10	13.207 (36.565)	0.841 *	13.194 (21.003)	0.663 ***	17.334 (48.537)	0.630 ***
	3	5	25.407 (21.711)	0.776 **	18.305 (37.674)	0.827 *	17.342 (31.368)	0.787 **
		10	17.311 (29.299)	0.814 *	13.912 (11.409)	0.711 ***	17.344 (19.11)	0.760 **
	5	5	20.436 (16.797)	0.852 *	14.721 (43.941)	0.550 ***	13.2 (76.801)	0.774 **
		10	13.199 (7.834)	0.652 ***	7.648 (6.348)	0.682 ***	12.498 (18.547)	0.570 ***
	6	5	16.428 (18.26)	0.703 ***	17.345 (37.083)	0.826 *	25.366 (69.292)	0.623 ***
		10	33.386 (36.713)	0.857 *	11.202 (15.079)	0.505 ***	26.836 (61.565)	0.496 ***
	7	5	17.345 (8.267)	0.674 ***	16.389 (16.73)	0.761 **	13.918 (32.05)	0.835 *
		10	20.433 (27.445)	0.908 n.s.	11.815 (9.285)	0.700 ***	28.331 (35.637)	0.500 ***

**Table 2 sensors-25-07396-t002:** Intraclass Correlation of PI and PF across trials and channels.

Variable	Channel	ICC	*F*	df1	df2	*p*	CI95%	Interpretation Based on [28]
Peak intensity (m/s^2^)	1	0.683	7.478	25	50	<0.0001	[0.49, 0.83]	Moderate agreement
	2	0.015	1.046	25	50	0.433	[−0.18, 0.28]	Poor agreement
	3	0.432	3.284	25	50	<0.001	[0.19, 0.66]	Poor agreement
	5	0.354	2.642	25	50	<0.001	[0.11, 0.6]	Poor agreement
	6	0.260	2.052	25	50	<0.05	[0.02, 0.52]	Poor agreement
	7	0.467	3.632	25	50	<0.0001	[0.23, 0.69]	Poor agreement
Peak frequency (Hz)	1	0.216	1.826	25	50	<0.05	[−0.02, 0.48]	Poor agreement
	2	0.074	1.240	25	50	0.254	[−0.13, 0.34]	Poor agreement
	3	0.341	2.553	25	50	<0.01	[0.1, 0.59]	Poor agreement
	5	0.136	1.472	25	50	0.121	[−0.08, 0.41]	Poor agreement
	6	−0.011	0.968	25	50	0.521	[−0.2, 0.25]	Poor agreement
	7	0.101	1.337	25	50	0.188	[−0.11, 0.37]	Poor agreement

**Table 3 sensors-25-07396-t003:** Gross, ipsilateral, and contralateral agreement of PI and PF across channels within the same trial.

Variable	Comparison	ICC	*F*	df1	df2	*p*	CI95%	Interpretation Based on [28]
**peak intensity**	Gross (all channels)	0.106	1.717	77	385	<0.0001	[0.04, 0.2]	Poor agreement
	Left channels	0.069	1.221	77	154	0.149	[−0.06, 0.22]	Poor agreement
	Right channels	0.097	1.331	77	154	0.068	[−0.03, 0.24]	Poor agreement
	1 vs. 5	0.521	3.229	77	77	<0.0001	[0.34, 0.66]	Moderate agreement
	2 vs. 6	0.140	1.322	77	77	0.111	[−0.09, 0.35]	Poor agreement
	3 vs. 7	0.617	4.496	77	77	<0.0001	[0.45, 0.74]	Moderate agreement
**peak frequency**	Gross (all channels)	0.097	1.655	77	385	<0.01	[0.03, 0.19]	Poor agreement
	Left channels	0.108	1.377	77	154	<0.05	[−0.02, 0.25]	Poor agreement
	Right channels	0.127	1.441	77	154	<0.05	[0.00, 0.28]	Poor agreement
	1 vs. 5	0.209	1.552	77	77	<0.05	[0.00, 0.41]	Poor agreement
	2 vs. 6	−0.053	.900	77	77	0.678	[−0.27, 0.17]	Poor agreement
	3 vs. 7	0.064	1.135	77	77	0.029	[−0.16, 0.28]	Poor agreement

**Table 4 sensors-25-07396-t004:** Tabulated comparison of inter-channel agreement and correlation of peak values and their respective peak time.

		Agreement					Correlation	
Channels	Variable	ICC	*F*	df1	df2	*p*	*τ*	*p*
1 vs. 5	Peak intensity	0.521	3.229	77	77	<0.001	0.584	<0.001
	Peak intensity time	0.972	70.327	77	77	<0.001	0.864	<0.001
	Peak frequency	0.209	1.552	77	77	<0.05	0.331	<0.001
	Peak frequency time	0.979	95.385	77	77	<0.001	0.889	<0.001
2 vs. 6	Peak intensity	0.140	1.322	77	77	0.111	0.279	<0.001
	Peak intensity time	0.952	40.406	77	77	<0.001	0.834	<0.001
	Peak frequency	−0.053	0.9	77	77	0.678	0.144	0.062
	Peak frequency time	0.964	54.638	77	77	<0.001	0.885	<0.001
3 vs. 7	Peak intensity	0.617	4.496	77	77	<0.001	0.420	<0.001
	Peak intensity time	0.942	33.603	77	77	<0.001	0.808	<0.001
	Peak frequency	0.064	1.135	77	77	0.289	0.277	<0.001
	Peak frequency time	0.993	307.928	77	77	<0.001	0.928	<0.001

## Data Availability

Supporting data can be provided by contacting the corresponding author.

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
