# Peer review of "Conformal Swallowing Accelerometry: Reimagining the Acquisition and Characterization of Swallowing Mechano-Acoustic Signals"

_sensors, 2025, doi:10.3390/s25237396_

Round 1
Reviewer 1 Report
Comments and Suggestions for Authors
Major Comments
- Statistical inconsistency in descriptive data
Table 1 presents both mean ± standard deviation and median [IQR] for the same variables. These represent distinct statistical assumptions (parametric vs. nonparametric). The authors should first assess data normality and then report either mean ± SD or median [IQR], according to the distribution of each variable. Combining both measures for the same dataset is inappropriate and should be corrected. - References within the abstract
The abstract includes numbered citations. According to MDPI and Sensors author guidelines, references are not allowed in the abstract section. The authors should remove all reference numbers or rephrase sentences in a general way. - Small and homogeneous sample
The study includes only 13 healthy young adults. This narrow sample restricts generalizability and external validity. The discussion should explicitly address this limitation and outline plans for including broader or clinical populations in future research. - Lack of physiological validation
No imaging or physiological reference (e.g., videofluoroscopy or FEES) was used to verify the origin of recorded signals. - Partial analysis of available channels
Although eight accelerometers were implemented, only six were analyzed. The rationale for excluding the inferior cricoid channel—recognized in the literature as an optimal recording site—should be clearly justified or incorporated into the analysis. - Statistical analysis depth
The paper relies mainly on nonparametric tests and ICCs. Complementary approaches such as linear mixed-effects models or multivariate time–frequency analyses would provide more robust insights into within-subject correlations and channel interactions. - Interpretation and tone
Some statements in the Discussion (e.g., “rediscovering the physiological process”) overstate the findings. The authors should adopt a more cautious tone, emphasizing that the results provide preliminary rather than definitive evidence.
Minor Comments
- The abstract should include key quantitative outcomes to strengthen its informativeness.
- Clarify the procedure for sample size determination and whether a power analysis was performed.
- Figures (especially Bland–Altman plots and boxplots) could be reorganized into composite panels for better readability.
- The manuscript is well written but excessively long. Shortening the Results and Discussion would improve clarity.
- The mention of a related patent requires an explicit conflict-of-interest statement.
- Ensure uniformity in units, decimal precision, and significant digits across all tables.
Author Response
Thank you very much for your time and valuable comments. Please see the PDF attachment for our response.

Reviewer 2 Report
Comments and Suggestions for Authors
The article “Conformal Swallowing Accelerometry: reimagining the acquisition and characterization of swallowing mechano-acoustic signals” by LAM et al. demonstrates a conformal accelerometer array to capture and analyze swallowing signals. While the results presented in the paper are very interesting, a fundamental issue needs to be addressed. The study utilized two different volumes of water (5mL and 10mL), but the paper lacks sufficient comparison and discussion of how these volumes differ in their swallowing characteristics. It’s a very important controlled variable to compare because it may directly affect swallowing biomechanics. Instead of comparing between trials, the authors should consider comparing between volumes. The author must justify these concerns before further consideration. Other comments as below
- What’s the possible cause of the erroneous spikes? And Figure 2 needs to add legends of the curves.
- It’s suggested to add the Roman numerals of the site defined in 2.4 to Figure 3 to help understand the placement.
- Line 202, is there a sensor placed on the shoulder? It’s neither mentioned afterwards nor shown in the figure.
- I don’t see the necessity of including the Morlet wavelet (Figure 5) in the article.
- Figure 6 is confusing and misleading, with a shared y-axis for acceleration and frequency.
- Trial 8 is very confusing and hard to follow. Again, I don’t think it’s a good idea to combine PI and PF into the same figure.
Author Response
Thank you for your time and valuable comments. Please see the PDF attachment for our response.

Round 2
Reviewer 1 Report
Comments and Suggestions for Authors
I consider all the modifications made to be sufficient and the article is ready for publication.
Author Response
Thank you very much for your time and valuable comments.
Reviewer 2 Report
Comments and Suggestions for Authors
In the revision of the article “Conformal Swallowing Accelerometry: reimagining the acquisition and characterization of swallowing mechano-acoustic signals”, the authors have attempted to address the concerns in their previous submission. The authors updated the results in Figure 5 and separated the two volumes for PI and PF. Though the author acknowledged there is “additional effect from increased bolus volume” (Line 449), the rest of the results of the paper still failed to sufficiently separate the condition. The author must justify how volume affects results (PI, PT, consistency, and coherence, etc). A major revision is suggested at this moment. See other comments below.
- Line 162, the authors need to provide quantifiable support for the claim of “extremely low background noise level”.
- What’s the resolution/sensitivity of the accelerometer?
- Please clarify how the signal baseline is calibrated to zero when calculating peak intensity. Is the mean baseline value subtracted from the raw signal?
- Does the statistical analysis correct for multiple comparisons?
- There is no volume comparison result discussed for Figure 5
- Figure 6 requires clarification. Please specify what 'count' represents. Does it aggregate data from all trials across all subjects for both volumes (3 x13 x2)? If so, shouldn't the two volumes be presented separately to allow comparison?
- Again, for Figure 7,8,9,10, Table 2, and Table 3, does the volume variable affect the results?
- For the peak intensity time (PIT), how is the time 0 being defined?
Author Response
Thank you for the comments and suggestions. Please find our response in the attached file.
